# Usefulness of *Mycobacterium tuberculosis*-polymerase chain reaction with bronchial washing samples in predicting discontinuation of airborne infection isolation in patients hospitalized with suspected pulmonary tuberculosis

**Tae-Ok Kim[1,2], Young-Ok Na[1], Hwa Kyung Park[1], Jae-Kyeong Lee[1], Hyung-Joo Oh🅞[1], Bo Gun Kho[1], Ha-Young Park[1], Yong-Soo Kwon[1,2], Yu-Il Kim[1,2], Sung-Chul Lim[1,2], Hong-Joon Shin🅞[1,2]***

**1** Department of Internal Medicine, Chonnam National University Hospital, Gwangju, Republic of Korea,
**2** Chonnam National University Medical School, Gwangju, Republic of Korea

\* 99naussica@naver.com

## Abstract

### Objective

In-hospital tuberculosis (TB) transmission remains a concern. Airborne infection isolation (AII) can be discontinued in hospitalized patients with suspected active pulmonary TB when the results of three consecutive sputum acid-fast bacilli (AFB) smears are negative. However, fiberoptic bronchoscopy can be performed in patients who may have difficulty in producing sputum samples. This study aimed to investigate the usefulness of *Mycobacterium tuberculosis*-polymerase chain reaction (MTB-PCR) with bronchial washing specimens in predicting AII discontinuation in hospitalized patients with suspected active pulmonary TB.

### Methods

We reviewed the medical charts of patients admitted to a tertiary hospital who were isolated and underwent fiberoptic bronchoscopy for suspicious pulmonary TB from January 2016 to December 2019. Patients with positive MTB-PCR results in the initial sputum examination were excluded. Criteria for discontinuing AII were defined as negative results for three consecutive AFB smears from respiratory specimens, or cases diagnosed other than TB. The study patients were divided into two groups: TB group and non-TB group.

### Results

In total, 166 patients were enrolled in the study. Of them, 35 patients were diagnosed with TB. There was no significant difference between the number of males in the TB (81; 61.8%) and non-TB (21; 60.0%) group. Though 139 patients had negative results on MTB-PCR using washing specimens, eight showed positive AFB culture. Of the 139 patients with

**Data Availability Statement:** All relevant data are within the paper and its Supporting Information files.

**Funding:** The authors report the following sources of funding: Grant BCRI20021 from Chonnam National University Hospital Biomedical Research Institute awarded to TOK and Grant 2022R1F1A1069623 from National Research Foundation of Korea funded by the Korean Government awarded to TOK. The funders had no role in study design, data collection and analysis, decision to publish, or preparation of the manuscript.

**Competing interests:** The authors have declared that no competing interests exist.

negative MTB-PCR results, 138 had negative results for three consecutive AFB smears or were established to not have pulmonary TB. Therefore, the predictive accuracy of MTB-PCR with bronchial washing samples for discontinuing AII was 99.2%.

## Conclusion

Although a negative result from MTB-PCR with bronchial washing samples cannot exclude pulmonary TB, it can predict AII discontinuation in hospitalized patients with suspected active pulmonary TB.

## Introduction

According to the 2020 World Health Organization tuberculosis (TB) report, 10.0 million patients were diagnosed with TB and 1.4 million people died from TB in 2019 [1]. As TB is transmitted through the air, airborne infection isolation (AII) is required for inpatients with suspected pulmonary TB. To diagnose pulmonary TB, acid-fast bacilli (AFB) smear examination, AFB culture, and *Mycobacterium tuberculosis*-polymerase chain reaction (MTB-PCR) using respiratory samples—such as expectorated or induced sputum, endotracheal aspirates, and fiberoptic bronchoscopy specimens—are required [2]. As it takes several weeks to confirm the results of AFB culture, the initial AFB smear and MTB-PCR results are important in diagnosing TB or determining the need for AII discontinuation. However, the criteria for discontinuing AII in patients who were not diagnosed with TB based on the initial AFB smear and MTB-PCR results have not been clearly established.

The Centers for Disease Control and Prevention (CDC) guidelines recommend that AII should be discontinued if the results of three consecutive AFB smear tests performed within an interval of 8 to 24 hours are negative [3]. However, delayed AII discontinuation was reported in 81% of patients with suspected TB [4]. Though a single sputum Xpert test significantly reduced the AII duration in a recent study, good-quality sputum specimens were needed for this test [5]. There was no study establishing the criteria for AII discontinuation in patients who have difficulty in producing good-quality sputum samples or who cannot undergo a sputum examination.

This study aimed to determine whether the results of an MTB-PCR from fiberoptic bronchoscopy specimens can predict AII discontinuation in hospitalized patients who are suspected of pulmonary TB.

## Patients and methods

### Study population

This study retrospectively investigated hospitalized patients in AII who underwent fiberoptic bronchoscopy for suspected pulmonary TB in a single tertiary hospital of South Korea from January 2017 to December 2019. All enrolled patients underwent fiberoptic bronchoscopy because they were unable to expectorate sputum or because TB could not be ruled out based on the results of initial sputum AFB smear and MTB-PCR. We screened patients undergoing fiberoptic bronchoscopy under AII during the study period. Patients who were already undergoing treatment for TB and those aged <18 years were excluded. Patients who were diagnosed by only clinical or histological findings were excluded. Patients who underwent fiberoptic bronchoscopy for purposes other than TB diagnosis were also excluded. TB was diagnosed

using the following methods: identification of *Mycobacterium tuberculosis* with culture or MTB-PCR. Patients were divided into two groups: the TB and non-TB groups. TB burden in South Korea is intermediate [6].

## Data collection

We reviewed the medical charts of the patients during the study period. The patients' age, sex, underlying diseases (hypertension, diabetes, previous TB history, and previous cerebral infarction history), and initial symptoms (cough, sputum, fever, dyspnea, and hemoptysis) were investigated. The results of AFB smear tests, MTB-PCRs, and TB culture using the sputum and fiberoptic bronchoscopy samples were collected. Chest computed tomography (CT) scans were also evaluated.

## Sputum examination

The sputum samples for AFB smear, AFB culture, and MTB-PCRs were placed in separate bottles and transferred to the laboratory. MTB-PCRs were performed using real-time MTB-PCR or Xpert MTB/RIF assay.

## Fiberoptic bronchoscopy procedures

Fiberoptic bronchoscopy was performed according to the guidelines [7]. Washing or bronchoalveolar lavage (BAL) was performed for the lesion with the highest TB risk according to the judgment of the test operator. In the case of multiple lesions, fiberoptic bronchoscopy samples were obtained from two sites at the discretion of the operator. All fiberoptic bronchoscopy samples were used for the AFB smear, AFB culture, and MTB-PCR. MTB-PCRs were performed using real-time MTB-PCR or Xpert MTB/RIF assay. Transbronchial lung biopsy was performed at the discretion of the operator.

## Radiologic findings

All enrolled participants underwent chest CT. Radiologic findings were assessed based on the radiologists' formal report or pulmonologists' judgment. Radiologic findings included nodules, cavities, consolidations, ground-glass opacities, bronchiectasis, and old TB lesions [8]. The involvement of the upper lobe was also investigated.

## Criteria for the discontinuation of AII

The criteria for the discontinuation of AII were negative results on a MTB-PCR using sputum or bronchial washing samples and negative results on three consecutive AFB smears using sputum or bronchial washing specimens or if TB was ruled out.

## Prediction of the discontinuation of AII

The predictive accuracy for discontinuation of AII of MTB-PCRs using bronchial washing samples was calculated as follows:

A: Numbers of patients with three consecutive negative AFB smear test results or diagnosed other than TB

B: Number of patients with negative MTB-PCR results using bronchial washing samples

$$\text{Predictive accuracy for possible discontinuation of AII} = \frac{A}{B} \times 100 \ (\%)$$

### Ethics statement

The Institutional Review Board at Chonnam National University Hospital (Gwangju, Republic of Korea) approved the study protocol and permitted the review and publication of our findings, as well as that of information derived from patient records (CNUH-01021-270). The requirement for informed consent was waived because of the retrospective nature of the study, and approved by ethic committee. Patient information was fully rendered innominate before analysis.

### Statistical analysis

All data were expressed as the median (interquartile range [IQR]) or number (percentage). Demographic and clinical variables were compared between the TB group and non-TB group using the chi-square test (for categorical variables) or Mann–Whitney U test (for continuous variables). Diagnostic performance in terms of sensitivity, specificity, positive predictive value, and negative predictive value was determined using TB culture as the gold standard.

All statistical analyses were performed using SPSS version 23.0 (IBM, Armonk, NY, USA); a P value of $<0.05$ was considered significant.

### Results

In total, 216 patients underwent isolation and fiberoptic bronchoscopy during the study period (Fig 1). Of them, 50 were excluded from the study owing to the following reasons: 16 were already undergoing TB treatment, 10 underwent fiberoptic bronchoscopy for the diagnosis of diseases other than TB, 8 showed positive results for a previous sputum MTB-PCR, 7 were isolated owing to influenza, 5 were aged $<18$ years, and 4 were diagnosed with TB by clinical or histologic findings. Finally, a total of 166 patients were enrolled. Of them, 35 patients were diagnosed with TB.

### Baseline characteristics in the pulmonary TB and non-TB groups

In the TB group, 35 were diagnosed with TB via MTB-PCR using bronchial washing specimens and TB culture (Table 1). Those in the non-TB group had the following diagnosis: pneumonia (46.6%), nontuberculous mycobacterial pulmonary disease (18.3%), and lung cancer (10.7%). There was no significant difference between the number of males in the TB (81; 61.8%) and non-TB (21; 60.0%) group. No significant difference was observed in the median age between the two groups (63.2 years in the non-TB group vs. 74.0 in the TB group; P = 0.132). A history of TB infection was more common in the non-TB group than in the TB group (20.6% vs. 57%; P = 0.045). Patients in the non-TB group had significantly more hemoptysis than those in the TB group. Of 166 patients, 9 (non-TB group: 6; TB group: 3) were not tested for sputum AFB smears, 31 (non-TB group: 20; TB group: 11) did not undergo the sputum MTB-PCR tests, and 8 (non-TB group: 4; TB group: 4) did not perform the sputum TB cultures. The number of patients with positive sputum AFB smear results was significantly higher in the TB group than in the non-TB group (17.1% vs. 2.4%; P = 0.001). Positive culture results were observed for 21 patients in sputum and for 26 patients in washing samples. Of these 17 patients had positive results both in sputum and washing samples. Thus, total 30 patients had positive cultures either in sputum or washing samples. Of 166 patients who were not diagnosed based on the results of the initial sputum examination, 27 were diagnosed via MTB-PCR using bronchial washing specimens. Nodules (85.7% vs. 51.1%; P < 0.000) were more frequent in the TB group than in the non-TB group.

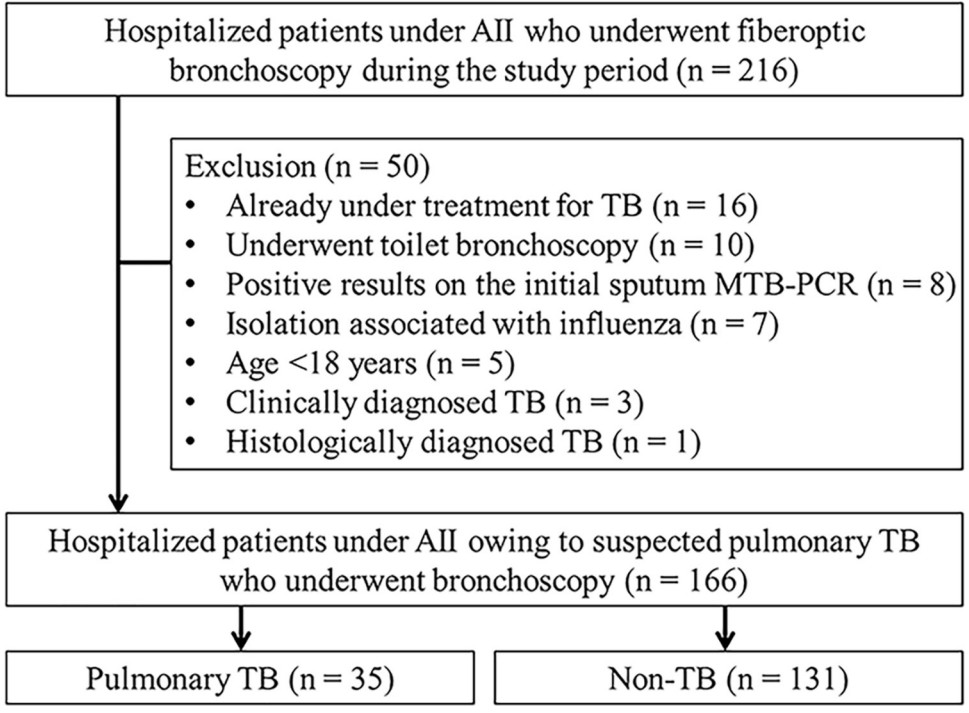

**Fig 1. Flowchart of study inclusion.** AII, airborne infection isolation; MTB, *Mycobacterium tuberculosis*; PCR, polymerase chain reaction.

### Diagnostic yield of MTB-PCR using washing samples

Of 27 patients who had positive results on MTB-PCR using washing specimens, 22 showed positive MTB culture. Of 139 patients with negative results on MTB-PCR using washing specimens, 8 showed positive MTB culture. Considering that MTB culture is the gold standard, MTB-PCR using washing specimens showed 73.3% sensitivity, 96.43% specificity, 81.4% positive predictive value, and 94.2% negative predictive value (Table 2).

There were 131 patients with negative results for both sputum AFB smears and sputum TB-PCR tests. Sixteen were positive in the MTB-PCR test using washing specimens and eighteen were positive for MTB culture. Of 16 patients who had positive results on MTB-PCR using washing specimens, 11 showed positive MTB culture. Of 115 patients with negative results on MTB-PCR using washing specimens, 7 showed positive MTB culture. MTB-PCR using washing specimens showed 61.1% sensitivity, 95.5% specificity, 68.7% positive predictive value, and 93.9% negative predictive value in patients with negative sputum AFB smear and MTB-PCR results (S1 Table).

### Prediction of the discontinuation of AII

Of 166 patients, 139 showed negative results for MTB-PCR using washing specimens (Table 3). Of the 139 patients with negative results on MTB-PCR using washing specimens, 8 were finally diagnosed with TB. Of them, 7 patients had negative results on three consecutive AFB smear tests. Of the 139 patients with negative MTB-PCR results, 138 had negative results for three consecutive AFB smears (n = 7) or were established not to have pulmonary TB (n = 131). Therefore, the predictive accuracy of MTB-PCR using washing specimens for the discontinuation of AII was 99.2% (Table 3).

**Table 1. Patients' baseline characteristics.**

| Variables | Non-TB (n = 131) | Pulmonary TB (n = 35) | P value |
|---|---|---|---|
| Age, years | 63.2 (52.0–76.0) | 74.0 (62.0–77.0) | 0.132 |
| Sex, male | 81 (61.8%) | 21 (60.0%) | 0.847 |
| Hypertension | 45 (34.4%) | 14 (40.0%) | 0.555 |
| Diabetes mellitus | 26 (19.8%) | 11 (31.4%) | 0.171 |
| Tuberculosis history | 27 (20.6%) | 2 (5.7%) | 0.045 |
| COPD | 10 (7.6%) | 1 (2.9%) | 0.461 |
| Old cerebrovascular disease | 9 (6.9%) | 4 (11.4%) | 0.476 |
| Initial symptoms | | | |
| Cough | 64 (48.9%) | 13 (37.1%) | 0.255 |
| Sputum | 55 (42.0%) | 9 (25.7%) | 0.117 |
| Fever | 35 (26.7%) | 9 (26.5%) | 1.000 |
| Dyspnea | 41 (31.3%) | 17 (48.6%) | 0.073 |
| Hemoptysis | 40 (30.5%) | 3 (8.6%) | 0.008 |
| Diagnostic evaluation | | | |
| Sputum AFB smear, positive | 3 (2.4%) | 6 (17.1%) | 0.001 |
| Sputum culture, positive | 0 | 21 (60.0%) | – |
| AFB smear using washing samples, positive | 4 (3.1%) | 8 (22.9%) | <0.000 |
| MTB-PCR using washing samples, positive | 0 | 27 (77.1%) | – |
| Culture using washing samples, positive | 0 | 26 (74.3%) | – |
| Radiologic findings | | | |
| Nodules | 67 (51.1%) | 30 (85.7%) | <0.000 |
| Consolidations | 95 (72.5%) | 21 (60.0%) | 0.213 |
| Cavitary diseases | 33 (25.2%) | 10 (28.6%) | 0.670 |
| Ground-glass opacities | 37 (28.2%) | 4 (11.4%) | 0.047 |
| Bronchiectasis | 22 (16.8%) | 3 (8.6%) | 0.293 |
| Old tuberculosis | 26 (19.8%) | 11 (31.4%) | 0.171 |
| Upper lobe involvement | 112 (85.5%) | 34 (97.1%) | 0.078 |
| Diseases other than TB | | | |
| Pneumonia | 61 (46.6%) | – | |
| NTM-lung disease | 24 (18.3%) | – | |
| Lung cancer | 14 (10.7%) | – | |

Data are presented as the median (interquartile range) or number (%).

MTB, *Mycobacterium tuberculosis*; COPD, chronic obstructive pulmonary disease; AFB, acid-fast bacilli; PCR, polymerase chain reaction; NTM, non-tuberculous mycobacterium.

**Table 2. Diagnostic yield of MTB-PCR using bronchoscopy samples considering AFB culture as the gold standard.**

| | Culture (+) | Culture (-) | Total | Sensitivity % | Specificity % | PPV % | NPV % |
|---|---|---|---|---|---|---|---|
| MTB-PCR (+) | 22 | 5 | 27 | 73.3 (54.1–87.7) | 96.3 (91.6–98.8) | 81.4 (64.4–91.4) | 94.2 (90.0–95.7) |
| MTB-PCR (-) | 8 | 131 | 139 | | | | |
| Total | 30 | 136 | 166 | | | | |

Data are presented as numbers. (+) positive; (–) negative.

MTB, *Mycobacterium tuberculosis*; PCR, polymerase chain reaction; AFB, acid-fast bacilli; PPV, positive predictive value; NPV, negative predictive value.

**Table 3. Prediction of the discontinuation of airborne infection isolation using bronchoscopy samples.**

| Variables | n | Accuracy |
|---|---|---|
| TB-PCR using washing samples, positive | 27 | – |
| TB-PCR using washing samples, negative | 139 | – |
| • TB culture, positive | 8 | – |
| : Three consecutive smear examinations, negative | 7 | – |
| : One smear examination, negative | 1 | |
| • Diseases other than TB | 131 | – |
| Predictability of AII discontinuation | – | 99.2% |

TB, tuberculosis; PCR, polymerase chain reaction; AII, airborne infection isolation.

## Subgroup analysis of TB group

Of the patients who showed negative results for MTB-PCR using sputum and washing samples, eight were diagnosed with TB. Age and sex were similar between the TB subgroups with negative (n = 8) and positive results (n = 27) on MTB-PCR using washing samples (Table 4).

**Table 4. Subgroup analysis of patients with tuberculosis.**

| Variables | Negative results on PCR using washing samples (n = 8) | Positive results on PCR using washing samples (n = 27) | P value |
|---|---|---|---|
| Age, years | 73.5 (65.7–77.0) | 74.0 (61.0–76.0) | 0.714 |
| Sex, male | 5 (62.5%) | 16 (59.3%) | 1.000 |
| Hypertension | 3 (37.5%) | 11 (40.7%) | 1.000 |
| Diabetes mellitus | 3 (37.5%) | 8 (29.6%) | 0.685 |
| TB history | 0 (0%) | 2 (7.4%) | 1.000 |
| COPD | 0 (0%) | 1 (3.7%) | 1.000 |
| Old cerebrovascular disease | 1 (12.5%) | 3 (11.1%) | 1.000 |
| Initial symptoms | | | |
| Cough | 4 (50.0%) | 9 (33.3%) | 0.433 |
| Sputum | 3 (37.5%) | 6 (22.2%) | 0.396 |
| Fever | 2 (28.6%) | 7 (25.9%) | 1.000 |
| Dyspnea | 5 (62.5%) | 12 (44.4%) | 0.443 |
| Hemoptysis | 1 (12.5%) | 2 (7.4%) | 0.553 |
| Diagnostic evaluation | | | |
| Sputum AFB smear, positive | 0 (0%) | 6 (25.0%) | 0.339 |
| Sputum culture, positive | 7 (87.5%) | 14 (58.3%) | 0.118 |
| AFB smear using washing samples, positive | 0 (0%) | 8 (29.6%) | 0.154 |
| Culture using washing samples, positive | 5 (62.5%) | 21 (77.8%) | 0.385 |
| Radiologic findings | | | |
| Nodules | 6 (75.0%) | 24 (88.9%) | 0.568 |
| Consolidations | 4 (50.0%) | 17 (63.0%) | 0.685 |
| Cavities | 2 (25.0%) | 8 (29.6%) | 1.000 |
| Ground-glass opacities | 0 (0%) | 4 (14.8%) | 0.553 |
| Bronchiectasis | 0 (0%) | 3 (11.1%) | 1.000 |
| Old tuberculosis | 1 (12.5%) | 10 (37.0%) | 0.387 |
| Upper lobe involvement | 8 (100.0%) | 26 (96.3%) | 1.000 |

Data are presented as the median (interquartile range) or number (%).

PCR, polymerase chain reaction; COPD, chronic obstructive pulmonary disease; AFB, acid-fast bacilli; MTB, *Mycobacterium tuberculosis*.

The TB subgroup with positive MTB-PCR results had a higher frequency of positive AFB smear results and positive TB culture using washing samples. However, no significant difference was observed in the findings between the two subgroups on imaging.

Five of the 27 patients with positive MTB-PCR results in washing samples had negative TB cultures (S2 Table). There were more ground-glass opacities on chest CT in culture-negative patients than in culture-positive patients (1 [4.5%] vs. 3 [60.0%]; P = 0.013).

## Discussion

This study investigated whether the results of MTB-PCRs using fiberoptic bronchoscopy samples could predict AII discontinuation in hospitalized patients who are suspected of pulmonary TB. Twenty-seven patients were diagnosed with TB by MTB-PCRs using fiberoptic bronchoscopy samples. Negative results for MTB-PCR using washing specimens were highly predictive of AII discontinuation. In patients diagnosed with TB, no significant clinical difference was observed between the groups with positive and negative results on MTB-PCRs using fiberoptic bronchoscopy samples.

To diagnose pulmonary TB, sputum specimens are required for sputum examinations, including AFB smear, AFB culture, and MTB-PCRs. Sputum induction or fiberoptic bronchoscopy is performed in patients experiencing difficulty with sputum production [9–11]. The diagnostic yield of sputum induction and fiberoptic bronchoscopy are similar [12–14]. In this study, none of the patients underwent sputum induction, and all patients underwent fiberoptic bronchoscopy. Twenty-seven patients, who were not diagnosed via sputum examination, were diagnosed with TB through MTB-PCRs using fiberoptic bronchoscopy samples. In this study, 139 patients had negative results on MTB-PCR using washing specimens; however, 8 showed positive AFB culture. Although the negative predictive value of MTB-PCR for diagnosis of TB was high at 94.2% in this study, negative results on MTB-PCR using fiberoptic bronchoscopy washing samples could not exclude pulmonary TB.

The risk factors for TB transmission include presence of cavitary disease, positive sputum MTB-PCR results, and positive AFB sputum smear results [15,16]. Pulmonary TB patients with positive AFB smear results have a higher risk of transmitting TB than those with negative AFB smear results [17,18]. The criteria for the discontinuation of AII with suspected pulmonary TB remain unclear. The CDC recommends the discontinuation of AII if the results of three serial sputum smear tests are negative [3]. Because the incremental diagnostic yield of a third sputum examination is low, a few hospitals consider two negative sputum smear test results as the criteria for the discontinuation of AII [19,20]. Recently, the US Food and Drug Administration stated that one or two negative sputum Xpert test results can be used as the criteria for discontinuing AII as an alternative to serial sputum AFB smears [21]. In a low-burden setting, negative sputum results of single or serial Xpert tests were useful for AII discontinuation and shortening the AII duration compared with serial sputum AFB smear results [5,11,20,22,23]. However, in patients who cannot produce enough sputum, one or two sputum AFB smear tests or MTB-PCRs may not be sufficient to determine the need for discontinuing AII. In this study, the predictability of MTB-PCRs using fiberoptic bronchoscopy samples for the discontinuation of AII was high in patients who are unable to expectorate sputum or those where TB could not be ruled out based on the results of initial sputum AFB smear and MTB-PCR.

The effectiveness of AII discontinuation remains controversial. TB transmission was observed in 13–20% of patients with smear-negative culture-positive TB [15,24,25]. Therefore, even if AII is discontinued in patients with negative results on three consecutive sputum AFB smear tests according to the CDC guidelines, TB cannot be excluded, and the risk of TB

transmission remains. However, additional studies are needed to determine whether culture-positive patients have a risk of TB transmission among patients with negative results on AFB smears and MTB-PCRs using fiberoptic bronchoscopy samples.

This study had several limitations. First, this was a retrospective study conducted in a single center. Therefore, there is a limit to the generalization of our findings. Second, four patients with pulmonary tuberculosis had positive sputum AFB smears and negative sputum MTB-PCR results. The sensitivity of sputum MTB-PCR is high if sputum AFB smears are positive in patients with pulmonary tuberculosis. Thus, these results might be due to errors in the sputum MTB-PCR test or insufficient sputum samples. Third, the effect of MTB-PCRs using fiberoptic bronchoscopy samples on the reduction of AII duration could not be evaluated in this study. Therefore, further prospective studies are needed to evaluate the effect of MTB-PCRs using fiberoptic bronchoscopy samples on reducing the duration of isolation.

## Conclusion

Although negative results on MTB-PCR using fiberoptic bronchoscopy washing samples cannot exclude pulmonary TB, the results can predict the discontinuation of AII in patients with suspected active pulmonary TB.

## Supporting information

**S1 Table. Diagnostic yield of MTB-PCR using bronchoscopy samples considering AFB culture as the gold standard in patients with negative sputum AFB smear and MTB-PCR results.**
(DOCX)

**S2 Table. Subgroup analysis of patients with positive MTB-PCR results.**
(DOCX)

**S1 File. Data set.**
(XLSX)

## Author Contributions

**Conceptualization:** Tae-Ok Kim, Hong-Joon Shin.

**Data curation:** Young-Ok Na, Hwa Kyung Park, Jae-Kyeong Lee, Hyung-Joo Oh, Bo Gun Kho, Ha-Young Park.

**Formal analysis:** Bo Gun Kho, Ha-Young Park, Yong-Soo Kwon, Yu-Il Kim, Sung-Chul Lim.

**Investigation:** Tae-Ok Kim, Sung-Chul Lim, Hong-Joon Shin.

**Methodology:** Yong-Soo Kwon, Yu-Il Kim, Sung-Chul Lim.

**Project administration:** Young-Ok Na, Hwa Kyung Park, Jae-Kyeong Lee, Hyung-Joo Oh.

**Resources:** Tae-Ok Kim, Hong-Joon Shin.

**Supervision:** Yong-Soo Kwon, Yu-Il Kim, Sung-Chul Lim, Hong-Joon Shin.

**Visualization:** Young-Ok Na, Hwa Kyung Park, Jae-Kyeong Lee, Hyung-Joo Oh, Bo Gun Kho, Ha-Young Park.

**Writing – original draft:** Tae-Ok Kim, Hong-Joon Shin.

**Writing – review & editing:** Tae-Ok Kim, Young-Ok Na, Hwa Kyung Park, Jae-Kyeong Lee, Hyung-Joo Oh, Bo Gun Kho, Ha-Young Park, Yong-Soo Kwon, Yu-Il Kim, Sung-Chul Lim, Hong-Joon Shin.

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
