## [Decision Letter · Decision Letter 0]

5 Jul 2022

PONE-D-21-35876Usefulness of Mycobacteriumtuberculosis-polymerase chain reaction with bronchial washing samples in predicting discontinuation of airborne infection isolation in patients hospitalized with suspected pulmonary tuberculosisPLOS ONE

Dear Dr. Shin,

Thank you for submitting your manuscript to PLOS ONE. After careful consideration, we feel that it has merit but does not fully meet PLOS ONE’s publication criteria as it currently stands. Therefore, we invite you to submit a revised version of the manuscript that addresses the points raised during the review process.

Please note that we have only been able to secure a single reviewer to assess your manuscript. We are issuing a decision on your manuscript at this point to prevent further delays in the evaluation of your manuscript. Please be aware that the editor who handles your revised manuscript might find it necessary to invite additional reviewers to assess this work once the revised manuscript is submitted. However, we will aim to proceed on the basis of this single review if possible. As you can see in the report attached below, the reviewer has raised some minor points about the methodology and result presentation. Please revise the manuscript to carefully address all the concerns raised.

We look forward to receiving your revised manuscript.

Kind regards,

Dario Ummarino, PhD

Senior Editor

PLOS ONE

Journal Requirements:

“This study was supported by the National Research Foundation of Korea funded by the Korean Government (grant 2019R1F1A1060899).”

“HJS

grant 2019R1F1A1060899

National Research Foundation of Korea funded by the Korean Government

he funders had no role in study design, data collection and analysis, decision to publish, or preparation of the manuscript.”

3.Please review your reference list to ensure that it is complete and correct. If you have cited papers that have been retracted, please include the rationale for doing so in the manuscript text, or remove these references and replace them with relevant current references. Any changes to the reference list should be mentioned in the rebuttal letter that accompanies your revised manuscript. If you need to cite a retracted article, indicate the article’s retracted status in the References list and also include a citation and full reference for the retraction notice.

Reviewers' comments:

Reviewer's Responses to Questions

**Comments to the Author**

1. Is the manuscript technically sound, and do the data support the conclusions?

Reviewer #1: Yes

2. Has the statistical analysis been performed appropriately and rigorously? 

Reviewer #1: Yes

3. Have the authors made all data underlying the findings in their manuscript fully available?

Reviewer #1: Yes

4. Is the manuscript presented in an intelligible fashion and written in standard English?

Reviewer #1: Yes

5. Review Comments to the Author

Reviewer #1: Lines 73 and 74. In the exclusion criteria, I understand that patients who were undergoing treatment for tuberculosis were excluded from the study. But why patients who were only diagnosed clinically or histologically were excluded?

Line 160. Where it says Of 166 patients, 11 (non-TB group: 6; TB group: 3) it should say Of 166 patients, 9 (non-TB group: 6; TB group: 3)

Why are in table 1 26 positive cultures for M.tuberculosis and in table 2 are 30 positive cultures?

Line 176. Where it says Of 136 patients with negative results it should say Of 139 patients with negative results

In lines 181 and 182 the name of table 2 instead of being

Table 2. Diagnostic yield of MTB-PCR using bronchoscopy samples considering AFB culture as the gold standard should be Table 2. Diagnostic yield of MTB-PCR using bronchoscopy samples considering AFB culture as the gold standard

In line 191 where it says A total of 139 patients were negative for TB it should say A total of 131 patients were negative for TB

In line 192 If we take into account that there were 131 negative cultures for M.tuberculosis and 139 negative MTB-PCR, the predictive accuracy would be 94.2%.

The authors explain very well the characteristics of the 8 patients who showed negative results for MTB-PCR using sputum and washing samples and positive culture for M.tuberculosis. But it is also important to explain the characteristics of the 5 patients who showed a positive MTB-PCR result and a negative culture for M.tuberculosis.

6. PLOS authors have the option to publish the peer review history of their article (what does this mean?). If published, this will include your full peer review and any attached files.

Reviewer #1: No

---

## [Author Response · Author response to Decision Letter 0]

23 Aug 2022

Responses to reviewers

Reviewer #1

Q1. Lines 73 and 74. In the exclusion criteria, I understand that patients who were undergoing treatment for tuberculosis were excluded from the study. But why patients who were only diagnosed clinically or histologically were excluded? 

A1. Thank you for your comments. 

Three patients with clinically diagnosed tuberculosis were excluded from this study. Despite being clinically diagnosed with TB, there was a limitation in evaluating the adequacy of air isolation in patients who showed negative results in all tests, including bronchoscopy.

One patient with histologically diagnosed TB was also excluded from this study. TB-PCR, TB-culture, and AFB/S tests were negative in the patient. Histologically, only granulation tissue was observed, but tissue TB-PCR was negative. Because it was unclear whether the patient had tuberculosis, we excluded the patient from this study.

Q2. Line 160. Where it says Of 166 patients, 11 (non-TB group: 6; TB group: 3) it should say Of 166 patients, 9 (non-TB group: 6; TB group: 3)

A2. Thank you for your comments. We changed the manuscript as your recommendation. [See line 160 on page 9]

Q3. Why are in table 1 26 positive cultures for M. tuberculosis and in table 2 are 30 positive cultures?

A3. Thank you for your comment. As shown in Table 1, positive culture results were observed for 21 patients in sputum and for 26 patients in washing samples. Of these 17 patients had positive results both in sputum and washing samples. Thus, total 30 patients had positive cultures either in sputum or washing samples. We modified the manuscript of results section. [See line 164-167 on page 10]

Q4. Line 176. Where it says Of 136 patients with negative results it should say Of 139 patients with negative results

A4. Thank you for your valuable comment. We changed the manuscript as your comment. [See line 194 on page 12] 

Q5. In lines 181 and 182 the name of table 2 instead of being

Table 2. Diagnostic yield of MTB-PCR using bronchoscopy samples considering AFB culture as the gold standard should be Table 2. Diagnostic yield of MTB-PCR using bronchoscopy samples considering AFB culture as the gold standard.

A5. The suggested sentence is the same as the existing sentence of title of table 2. [See line 184-185 on page 11]

Q6. In line 191 where it says A total of 139 patients were negative for TB it should say A total of 131 patients were negative for TB

A6. Thank you for your valuable comment. This sentence was deleted because its meaning was unclear. Instead, we provided a detailed description of predictive accuracy. [See line 194-195 on page 12]

Q7. In line 192 If we take into account that there were 131 negative cultures for M.tuberculosis and 139 negative MTB-PCR, the predictive accuracy would be 94.2%.

A7. Thank you for your valuable comment. As we mentioned in the method section, the prediction of AII discontinuation is calculated as the percentage of patients with three consecutive negative AFB smear test results (n = 7) or were established not to have pulmonary TB (n = 131) out of patients with negative MTB-PCR results using bronchial washing samples (n = 139) [(7+131)/139×100 (%)]. Therefore, the prediction of the discontinuation of AII is 99.2%. We also modified abstract in the result section. [See line 194-195 on page 12]

Q8. The authors explain very well the characteristics of the 8 patients who showed negative results for MTB-PCR using sputum and washing samples and positive culture for M.tuberculosis. But it is also important to explain the characteristics of the 5 patients who showed a positive MTB-PCR result and a negative culture for M.tuberculosis.

A8. Thank you for your valuable comments. A subgroup analysis was conducted for 27 patients with positive MTB-PCR results as your recommendation. Five of the 27 patients with positive MTB-PCR results in washing samples had negative TB cultures. There were more ground-glass opacities on chest CT in culture-negative patients than in culture-positive patients (1 [4.5%] vs. 3 [60.0%]; P = 0.013). There were no significant differences between the two groups in terms of other clinical variables. These results have been added to Table S1 and the results section. [See line 216-218 on page 14]

---

## [Decision Letter · Decision Letter 1]

25 Oct 2022

PONE-D-21-35876R1Usefulness of Mycobacteriumtuberculosis-polymerase chain reaction with bronchial washing samples in predicting discontinuation of airborne infection isolation in patients hospitalized with suspected pulmonary tuberculosisPLOS ONE

Dear Dr. Shin,

Thank you for submitting your manuscript to PLOS ONE. After careful consideration, we feel that it has merit but does not fully meet PLOS ONE’s publication criteria as it currently stands. Therefore, we invite you to submit a revised version of the manuscript that addresses the points raised during the review process.

We look forward to receiving your revised manuscript.

Kind regards,

Mao-Shui Wang

Academic Editor

PLOS ONE

Journal Requirements:

Reviewers' comments:

Reviewer's Responses to Questions

**Comments to the Author**

1. If the authors have adequately addressed your comments raised in a previous round of review and you feel that this manuscript is now acceptable for publication, you may indicate that here to bypass the “Comments to the Author” section, enter your conflict of interest statement in the “Confidential to Editor” section, and submit your "Accept" recommendation.

Reviewer #2: (No Response)

Reviewer #3: All comments have been addressed

2. Is the manuscript technically sound, and do the data support the conclusions?

Reviewer #2: Yes

Reviewer #3: Yes

3. Has the statistical analysis been performed appropriately and rigorously? 

Reviewer #2: Yes

Reviewer #3: Yes

4. Have the authors made all data underlying the findings in their manuscript fully available?

Reviewer #2: Yes

Reviewer #3: Yes

5. Is the manuscript presented in an intelligible fashion and written in standard English?

Reviewer #2: Yes

Reviewer #3: Yes

6. Review Comments to the Author

Reviewer #2: I have read the V1 and R1 version of the article and the other reviewer’s comment to V1. This is my first revision of the article. I have some additional comments that in my opinion may improve the quality of the paper.

- I think it would be of interest to know how many patients with suspected TB were included because of AFB and MTB-PCR negative results in sputum smear, and how many because they could not expectorate. Presenting data of patients with MTB-PCR negative in sputum, but positive only after bronchoscopy would be of interest, indicating the sensitivity of MTB-PCR on bronchoscopy in patients with negative results in sputum.

- It is somehow surprising the six patients with positive AFB in smear were finally diagnosed of pulmonary TB only after bronchoscopy. Was MTB-PCR performed on the positive AFB sputum sample and were in all the six patients negative? (MTB-PCR in AFB+ sputum samples has a very high sensitivity close to 100%)

Other minor comments:

- It is difficult for me to understand inclusion criteria of the initial review of 216 patients:

Fig-1 and page 9, lines 140-145. The authors state that inclusion criteria were “patients under AII who underwent fiberoptic bronchoscopy for pulmonary TB, because they were unable to expectorate or because TB could not be ruled out based on the results of initial AFB smear and MTB-PCR”. So it is difficult for me to understand why among the initial 216 patients (“hospitalized patients under AII owing to suspected pulmonary TB who underwent bronchoscopy”) are included 16 patients already under treatment for TB; 8 patients with positive MTB-PCR in sputum or 3 patients with clinically diagnosed TB (they were already with a TB dg at the time performing the bronchoscopy).

- Page 4. Line 57. Interval 8 to 24 hours (not just 8h) according to ref #3.

- Although I’m not an English native speaker; I think there are some grammatical errors: page 6; line-109: results of a MTB-PCR; line: 111: if TB was ruled out...

Reviewer #3: (No Response)

7. PLOS authors have the option to publish the peer review history of their article (what does this mean?). If published, this will include your full peer review and any attached files.

Reviewer #2: No

Reviewer #3: No

---

## [Author Response · Author response to Decision Letter 1]

20 Nov 2022

Reviewer #2: I have read the V1 and R1 version of the article and the other reviewer’s comment to V1. This is my first revision of the article. I have some additional comments that in my opinion may improve the quality of the paper.

Q1) I think it would be of interest to know how many patients with suspected TB were included because of AFB and MTB-PCR negative results in sputum smear, and how many because they could not expectorate. Presenting data of patients with MTB-PCR negative in sputum, but positive only after bronchoscopy would be of interest, indicating the sensitivity of MTB-PCR on bronchoscopy in patients with negative results in sputum.

A1) Thank you for your comments. There were 131 patients with negative results for both sputum AFB smears and sputum TB-PCR tests. Sixteen were positive in the MTB-PCR test using washing specimens and eighteen were positive for MTB culture. Of 16 patients who had positive results on MTB-PCR using washing specimens, 11 showed positive MTB culture. Of 115 patients with negative results on MTB-PCR using washing specimens, 7 showed positive MTB culture. MTB-PCR using washing specimens showed 61.1% sensitivity, 95.5% specificity, 68.7% positive predictive value, and 93.9% negative predictive value in patients with negative sputum AFB smear and MTB-PCR results. We presented in table S1. (See line 191-198 on page 12)

Q2) It is somehow surprising the six patients with positive AFB in smear were finally diagnosed of pulmonary TB only after bronchoscopy. Was MTB-PCR performed on the positive AFB sputum sample and were in all the six patients negative? (MTB-PCR in AFB+ sputum samples has a very high sensitivity close to 100%)

A2) Thank you for your comments. The sputum AFB smear test was positive in nine of the patients enrolled in this study. Sputum TB-PCR was performed by six of them, but all of them were negative. AFB smears and TB-PCR results were positive in four of these six patients. NTM was cultured for the other two patients. AFB smear and TB-PCR tests performed on washing samples were positive for two out of three patients who did not undergo sputum TB-PCR. All tests on the remaining patient's washing specimen were negative. Sputum MTB-PCR has a very high sensitivity if AFB smear is positive in a sputum sample of a patient with pulmonary tuberculosis, as you mentioned in your comments. There were four patients in this study who had positive sputum AFB smears, but negative sputum MTB-PCRs, and tuberculosis was diagnosed through washing samples. There may have been errors in the sputum MTB-PCR tests or insufficient samples of sputum MTB-PCR have been collected. We added this limitation in the limitation section. (See line 274-278 on page 16-17)

Other minor comments:

Q3) It is difficult for me to understand inclusion criteria of the initial review of 216 patients:

Fig-1 and page 9, lines 140-145. The authors state that inclusion criteria were “patients under AII who underwent fiberoptic bronchoscopy for pulmonary TB, because they were unable to expectorate or because TB could not be ruled out based on the results of initial AFB smear and MTB-PCR”. So it is difficult for me to understand why among the initial 216 patients (“hospitalized patients under AII owing to suspected pulmonary TB who underwent bronchoscopy”) are included 16 patients already under treatment for TB; 8 patients with positive MTB-PCR in sputum or 3 patients with clinically diagnosed TB (they were already with a TB dg at the time performing the bronchoscopy).

A3) Thank you for your valuable comments. To identify patients under AII who underwent fiberoptic bronchoscopy for pulmonary TB, because they were unable to expectorate or because TB could not be ruled out based on the results of initial AFB smear and MTB-PCR, the following patients were screened at first: patients under AII who underwent fiberoptic bronchoscopy during the study period. 216 patients were screened, and 50 were excluded based on the exclusion criteria. To clarify the meaning of figure 1, we changed it. (See line 72-73 on page 5 and figure 1)

Q4) Page 4. Line 57. Interval 8 to 24 hours (not just 8h) according to ref #3.

A4) Thank you for your valuable comment. We change the manuscript as your recommendation. (See line 57 on page 4)

Q5) Although I’m not an English native speaker; I think there are some grammatical errors: page 6; line-109: results of a MTB-PCR; line: 111: if TB was ruled out...

A5) Thank you for your comment. We changed the manuscript as your comment. (See line 110 and 112 on page 6)

---

## [Decision Letter · Decision Letter 2]

4 Dec 2022

Usefulness of Mycobacteriumtuberculosis-polymerase chain reaction with bronchial washing samples in predicting discontinuation of airborne infection isolation in patients hospitalized with suspected pulmonary tuberculosis

PONE-D-21-35876R2

Dear Dr. Shin,

We’re pleased to inform you that your manuscript has been judged scientifically suitable for publication and will be formally accepted for publication once it meets all outstanding technical requirements.

Kind regards,

Mao-Shui Wang

Academic Editor

PLOS ONE

Additional Editor Comments (optional):

Reviewers' comments:

Reviewer's Responses to Questions

**Comments to the Author**

1. If the authors have adequately addressed your comments raised in a previous round of review and you feel that this manuscript is now acceptable for publication, you may indicate that here to bypass the “Comments to the Author” section, enter your conflict of interest statement in the “Confidential to Editor” section, and submit your "Accept" recommendation.

Reviewer #2: All comments have been addressed

2. Is the manuscript technically sound, and do the data support the conclusions?

Reviewer #2: (No Response)

3. Has the statistical analysis been performed appropriately and rigorously? 

Reviewer #2: (No Response)

4. Have the authors made all data underlying the findings in their manuscript fully available?

Reviewer #2: (No Response)

5. Is the manuscript presented in an intelligible fashion and written in standard English?

Reviewer #2: (No Response)

6. Review Comments to the Author

Reviewer #2: In my opinion, the authors have correctly addressed the comments and I have no further recommendations.

7. PLOS authors have the option to publish the peer review history of their article (what does this mean?). If published, this will include your full peer review and any attached files.

Reviewer #2: No

---

## [Editor Report · Acceptance letter]

19 Dec 2022

PONE-D-21-35876R2 

Usefulness of *Mycobacterium tuberculosis*-polymerase chain reaction with bronchial washing samples in predicting discontinuation of airborne infection isolation in patients hospitalized with suspected pulmonary tuberculosis 

Dear Dr. Shin:

I'm pleased to inform you that your manuscript has been deemed suitable for publication in PLOS ONE. Congratulations! Your manuscript is now with our production department. 

Kind regards, 

on behalf of

Dr. Mao-Shui Wang 

Academic Editor

PLOS ONE